# Self-Disentanglement and Re-Composition for Cross-Domain Few-Shot Segmentation

**Jintao Tong** [1]   **Yixiong Zou**[✉ 1]   **Guangyao Chen** [2]   **Yuhua Li** [1]   **Ruixuan Li** [1]

## Abstract

Cross-Domain Few-Shot Segmentation (CD-FSS) aims to transfer knowledge from a source-domain dataset to unseen target-domain datasets with limited annotations. Current methods typically compare the distance between training and testing samples for mask prediction. However, we find an entanglement problem exists in this widely adopted method, which tends to bind source-domain patterns together and make each of them hard to transfer. In this paper, we aim to address this problem for the CD-FSS task. We first find a natural decomposition of the ViT structure, based on which we delve into the entanglement problem for an interpretation. We find the decomposed ViT components are crossly compared between images in distance calculation, where the rational comparisons are entangled with those meaningless ones by their equal importance, leading to the entanglement problem. Based on this interpretation, we further propose to address the entanglement problem by learning to weigh for all comparisons of ViT components, which learn disentangled features and re-compose them for the CD-FSS task, benefiting both the generalization and finetuning. Experiments show that our model outperforms the state-of-the-art CD-FSS method by 1.92% and 1.88% in average accuracy under 1-shot and 5-shot settings, respectively.

## 1. Introduction

Recent progress in deep neural networks (Long et al., 2015; Zhao et al., 2017; Dosovitskiy et al., 2020) has been driven by large-scale annotated datasets. However, the reliance

[1]School of Computer Science and Technology, Huazhong University of Science and Technology, Wuhan, China [2]School of Computer Science, Peking University, Beijing, China. Correspondence to: Yixiong Zou[✉] <yixiongz@hust.edu.cn>.

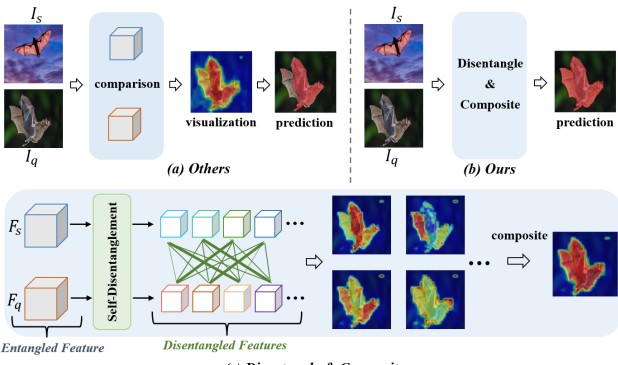

Figure 1: (a) A problem of feature entanglement exists in current works, which entangles multiple patterns and reduces the transferability. (b)(c) To handle this problem, we find a natural decomposition in ViT's feature, then analyze the entanglement problem based on this decomposition, and finally propose to self-disentangle and re-compose the ViT feature to address this problem for efficient cross-domain transferring and target-domain adaptation.

on abundant labeled data poses a major challenge, especially for dense prediction tasks like semantic segmentation. Cross-Domain Few-shot Semantic Segmentation (CD-FSS) (Shaban et al., 2017; Dong & Xing, 2018; Zhang et al., 2020b; Lei et al., 2022) has been introduced to address this issue, enabling predictions for target-domain unseen classes by limited annotated samples, with knowledge transferred from a data-sufficient source domain.

Existing CD-FSS works (Herzog, 2024; Su et al., 2024; He et al., 2024; Tong et al., 2024) usual perform segmentation by measuring the similarity between the support and query set based on features output by the encoder (Fig. 1a). However, we find this well-adopted method always leads to the entanglement of multiple patterns[1] and harms the transferability. For example, in Fig. 1a, the model tends to entangle the patterns of wings and bodies, i.e., detecting wings and bodies only when these two patterns appear simultaneously. However, if an image contains only the wings but the body is different from the training data (e.g., another kind of

---
[1]Represent various attributes such as object region, texture, color, and more, we visualize the activated object regions.

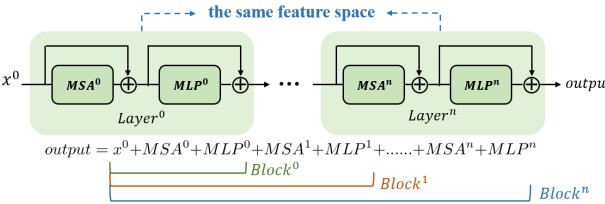

$$output = x^0 + MSA^0 + MLP^0 + MSA^1 + MLP^1 + \ldots + MSA^n + MLP^n$$

Figure 2: The residual connection and consistent spatial size make the output of ViT components located in the same feature space, which inspires us to view the final output of a ViT as the cumulative composition of all ViT components.

bat), the model may fail to capture the wings, leading to segmentation errors. For the CD-FSS task, domain gaps and semantic gaps heavily exist between the source and target datasets. Therefore, transferring entangled patterns is much more difficult than transferring disentangled ones. Inspired by this issue, in this paper, we aim to address the entanglement problem for the CD-FSS task (Fig. 1bc).

Recent work on ViT interpretability (Gandelsman et al.) shows that the residual connections and the consistent spatial size make the output of each ViT component (e.g., MSA, MLP) located in the same feature space. Therefore, we find the final output of a ViT can naturally be seen as a cumulative composition of all ViT components (shown in Fig.2 and detailed in Section 2.2). Such a structural decomposition of ViT's output inspires us to ask: ***can the entangled semantic patterns also be decomposed in this way?***

Based on this inspiration, we first delve into the entanglement problem for an interpretation. We find each ViT component captures distinct semantic patterns, e.g., bodies and wings, while the cumulative composition of these components implicitly binds all these patterns in ViT's output. By comparing distances between two images, as a mainstream of CD-FSS methods, the model essentially combines all possible comparisons between different components equally. Therefore, rational comparisons between patterns (wings vs. wings) are entangled with those meaningless comparisons (bodies vs. wings) by their equal importance, which we interpret to cause the feature entanglement problem.

Inspired by this interpretation, we further propose to handle the entanglement problem by learning to weigh for all comparisons between ViT components. Specifically, we first self-disentangle ViT's output by extracting features of different ViT components. Then, we introduce an Orthogonal Space Decoupling (OSD) module to further reduce the correlation of the disentangled features. Given these disentangled component features, we propose a Cross-Pattern Comparison (CPC) module, where the disentangled patterns are compared crossly for the re-composition, based on weights generated by OSD to emphasize the comparison between components with the same position. During the target-domain finetuning, we further introduce the Adaptive

Fusion Weight (AFW) to dynamically learn the comparison weights for efficient adaptation (Fig. 1b).

To sum up, our primary contributions are as follows:

- To our knowledge, we are the first to analyze the feature entanglement problem from the aspect of the natural decomposition of ViT structures for the CD-FSS task.
- We interpret the entanglement problem as a result of entangling rational comparisons between ViT components with those meaningless ones by their equal importance.
- Based on this interpretation, we further propose to self-disentangle and re-compose ViT components for the CD-FSS task with the proposed orthogonal space decoupling module, cross-comparison module, and adaptive fusion weight module, which addresses the entanglement problem by learning to weigh for each comparison, benefiting both the generalization and finetuning for CD-FSS.
- Extensive experiments show the effectiveness of our work on four different CD-FSS scenarios. Our model significantly outperforms state-of-the-art methods.

## 2. Delve into Feature Entanglement

In this section, we delve into the causes of feature entanglement by decomposing the structure of the ViT output.

### 2.1. Problem Definition

Cross-domain few-shot semantic segmentation (CD-FSS) aims to transfer knowledge learned from the source domain to unseen target domains with only a few annotated support images. Consider a source domain $D_s = (\mathcal{X}_s, \mathcal{Y}_s)$ and a target domain $D_t = (\mathcal{X}_t, \mathcal{Y}_t)$, where $\mathcal{X}$ denotes the input distribution and $\mathcal{Y}$ denotes the label space. The input data distributions of $D_s$ and $D_t$ are distinct, and their label spaces do not overlap, i.e., $\mathcal{X}_s \neq \mathcal{X}_t$, $\mathcal{Y}_s \cap \mathcal{Y}_t = \emptyset$. The model is trained solely on $D_s$ and without access to the target data, and then applied to segment novel classes in $D_t$.

In this work, we adopt the meta-learning episodic manner following (Lei et al., 2022) to train and test our model. Specifically, both the training set from $D_s$ and the testing set from $D_t$ consist of several episodes. Each episode includes $K$ support samples $S = \{I_s^i, M_s^i\}_{i=1}^K$ ($K$ image-mask pairs) and a query $Q = \{I_q, M_q\}$, where $I$ represents the image and $M$ denotes the label. Within each episode, the model is expected to use the support sample $\{I_s, M_s\}_{i=1}^K$ and the query image $I_q$ to predict the query label.

### 2.2. Structural Decomposition of the ViT Output

**ViT architecture.** ViT (Dosovitskiy et al., 2020) is a residual network built from $L$ layers, each of which contains a multi-head self-attention (MSA) followed by an MLP block. The input $I$ is first split into $N$ non-overlapping image patches. The patches are projected linearly into $N$ $d$-dimensional vectors, and positional embeddings are added

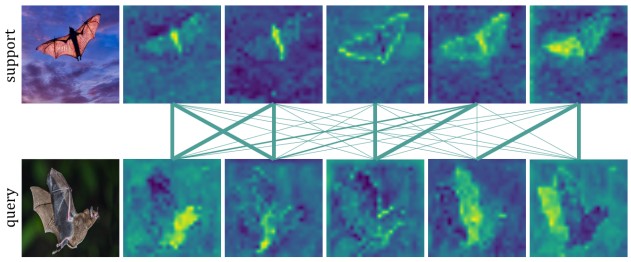

Figure 3: Visualization of the cross-match between layers, where the same column means the same layer ID and bold lines indicate rational matches.

to them to create the $image\ tokens\{z_i^0\}_{i\in\{1,...,N\}}$. Due to the segmentation task, the $CLS\ token$ is excluded (not included in the following formulas). Formally, the matrix $Z^0 \in \mathbb{R}^{d \times N}$, with the tokens $z_1^0, z_2^0, ..., z_N^0$ as columns, constitutes the initial state of the residual stream. It is updated for $L$ iterations via these two residual steps:

$$\hat{Z}_l = \text{MSA}^l(Z^{l-1}) + Z^{l-1}, \ Z_l = \text{MLP}^l(\hat{Z}^l) + \hat{Z}^l \quad (1)$$

**Decomposition of the ViT.** The residual structure of ViT allows us to express its output as a sum of the direct contributions of individual layers of the model. By unrolling Eq. 1 across layers, the image representation $\text{ViT}(I)$ can be written as (Both here and in Eq. 1, we ignore a layer-normalization term to simplify derivations):

$$\text{ViT}(I) = Z^0 + \sum_{l=1}^{L} \text{MSA}^l(Z^{l-1}) + \sum_{l=1}^{L} \text{MLP}^l(\hat{Z}^l) \quad (2)$$

We ignore here the indirect effects of the output of one layer on another downstream layer, and further simplify the MLPs and MSAs into Layers[2]:

$$\text{ViT}(I) = Z^0 + \sum_{l=1}^{L} \text{Layer}^l \quad (3)$$

**2.3. Analyzing Entanglement by ViT Decomposition**

Since most CD-FSS methods are based on distances between support and query set images, we begin our analysis by revisiting the distance comparison. For a support-query pair $\{I_s, I_q\}$, their features extracted by ViT are:

$$Z_s = \text{ViT}(I_s), \quad Z_q = \text{ViT}(I_q) \quad (4)$$

The similarity score $S$ is computed using cosine similarity:

$$S = Z_s \cdot Z_q / \|Z_s\| \|Z_q\| \quad (5)$$

Substituting Eq 3 and Eq 4 into the similarity formula:

$$S = \left( Z_s^0 + \sum_{l=1}^{L} \text{Layer}_s^l \right) \cdot \left( Z_q^0 + \sum_{l=1}^{L} \text{Layer}_q^l \right) / \|Z_s\| \|Z_q\|. \quad (6)$$

From Eq. 6, we can see a cross-match of different layers in the distance calculation:

$$\tilde{S} = \left(\sum_{i=1}^{L} \text{Layer}_s^i\right) \cdot \sum_{j=1}^{L} (\text{Layer}_q^j) = \sum_{i=1}^{L} \sum_{j=1}^{L} (\text{Layer}_s^i \cdot \text{Layer}_q^j) \quad (7)$$

---

[2]Please see Fig. 2 for distinguishing Layer and Block.

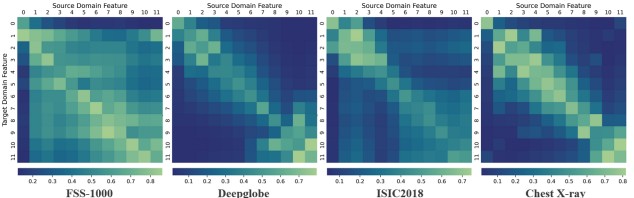

Figure 4: Domain similarities between source- and target-domain features extracted from different layers. A brighter color means a higher domain similarity, indicating less over-fitting to the source domain and less feature entanglement.

| Target Dataset | FSS-1000 | Deepglobe | ISIC | ChestX |
|---|---|---|---|---|
| Final Output | 0.4288 | 0.3135 | 0.2527 | 0.2856 |
| Layer-wise Avg. | 0.6107 | 0.4988 | 0.5074 | 0.6612 |
| Top-12 Avg. | 0.8126 | 0.6441 | 0.6164 | 0.7823 |
| Bottom-12 Avg. | 0.1407 | 0.0130 | 0.0473 | 0.0163 |

Table 1: Simply shifting cross-matched layers heavily affects domain similarities, inspiring us to handle the entanglement problem by learning the cross-match of layers.

This implies the output of every layer is compared with all other layers. Since all layers are in the same feature space (Fig. 2), it is feasible to compare even the output of the first layer and the last layer, although the comparison results may be meaningless (Fig.3). However, in Eq. 6, we observe the matching process treats all layers equally, which means even the meaningless comparison between two distinct layers will have a non-trivial impact on the final distance.

As the feature entanglement can be viewed as a kind of over-fitting to the source domain, which can be represented as the recognition based on meaningless patterns, such meaningless comparisons would lead the model to rely on patterns specific to such comparisons, leading to overfitting. However, Eq. 6 entangles these patterns and comparisons together with equally. Therefore, **we hypothesize it is the entanglement in the cross-match of different layers that leads to the entanglement in the semantic features**.

**Validation of hypothesis.** To validate this hypothesis, we use domain similarities between source and target domains to measure the feature entanglement, i.e., feature entanglement leads to overfitting to the source domain, and more overfitting leads to less transferable features across domains, reducing the domain similarity. We follow (Zou et al., 2024a) to take the CKA similarity[3] to measure the domain similarity. Specifically, we use different ViT layers to extract features from each domains, and then compare features from the source domain and target domains to measure the CKA similarity. Since ViT contains 12 layers, this would lead to $12 \times 12$ CKA values for each source-target domain pair. As shown in Fig. 4, the cross-match deviating from the diagonal shows much lower domain similarities, e.g., the

---

[3]Please refer to the appendix A for detailed formulations.

top-right corner which means matching Layer 0 of the target domain and Layer 11 of the source domain. In Table 1, we also calculate the CKA of the comparison between final outputs (i.e., viewed as the average of Fig. 4) and the layer-wise comparison (i.e., comparing outputs with the same layer ID, the diagonal of Fig. 4). We can see the layer-wise domain similarity is much higher than that of the final output. This verifies that the correct match between layers can lead to higher domain similarity, and therefore less feature entanglement. In other words, the decomposed components (Layers) are well-suited for generalization themselves. It is the cross-match of components that leads to feature entanglement.

Moreover, in Fig. 4, we can also observe a small fraction of cross-matches show higher CKA values than the diagonal (layer-wise) ones. This indicates the patterns captured by each layer are not strictly different from others (Fig. 3), possibly due to the dynamic calculation of the self-attention mechanism (Park & Kim, 2022). To verify it, in Table 1, we simply shift the match between layers, and the domain similarities are even higher than the layer-wise ones, indicating a learnable cross-match may be better than the naive layer-wise match for solving the entanglement problem.

## 2.4. Discussion and conclusion

To handle the feature entanglement problem for CD-FSS, we revisit ViT's inherent structure, which provides a natural decomposition of its features. Since all internal features of ViT layers (components) are in the same feature space due to the residual connection, ViT's final output implicitly combines all component features with the same importance. This also leads to the cross-match with equal importance between all ViT components. By taking the domain similarity as a measure of the source-domain overfitting caused by the entanglement problem, we find it is the meaningless cross-match between ViT components that majorly causes the entanglement problem, where the rational matches are entangled with those meaningless ones by their equal weights. Inspired by these, we aim to handle this problem by learning the cross-match weights of components.

## 3. Method

Building on our analysis of feature entanglement, we propose the concept of self-disentanglement and re-composition. Our framework is illustrated in Fig. 5. We first extract support and query features from different ViT layers, concatenate them along the channel dimension, and feed them into the Orthogonal Space Decoupling (OSD) module for weight allocation and semantic disentanglement. Subsequently, the outputs of OSD are input into the Cross-Pattern Comparison (CPC) module, where the disentangled patterns are compared crossly for the re-composition of patterns. For the re-composition, during source-domain training, score maps are composed with weights from OSD for efficient pat-

tern learning. During target-domain finetuning, the Adaptive Fusion Weight (AFW) is introduced to dynamically learn the comparison weights for efficient adaptation.

### 3.1. Orthogonal Space Decoupling

Since misaligned and correct matches are assigned equal weights during comparison, we need to rectify misaligned matches. This involves two steps: (1) adjusting the semantics of each feature and assigning different weights before comparison, and (2) allocating different weights to each similarity after comparison. Therefore, we propose the OSD module as an explicit global decoupling and weight allocation mechanism. It helps semantic disentanglement by aggregating feature channels, enforcing orthogonal constraints, and assigning appropriate weights to different patterns.

Specifically, given a support and query set, a sequence of $L$ pairs of support and query feature maps $\{(F_l^s, F_l^q)\}_{l=1}^L$ is extracted from various ViT layers. Each representation $F_l \in \mathbb{R}^{d \times N}$ is reshaped to $F_l \in \mathbb{R}^{d \times n \times n}$, where $n$ is the patch number and the $d$ is channel dimension. These support and query patterns are first concatenated along the channel dimension to form a complete representation:

$$F_{con}^* = concat(\{F_l^*\}_{l=1}^L) \qquad (8)$$

where $F_{con}^* \in \mathbb{R}^{Ld \times n \times n}$ and $*$ denotes that both the support and query patterns undergo the same operation.

Then, these concatenated features are fed into the OSD, where explicit constraints on each pattern channel enable semantic decoupling, while handling weight allocation. The OSD consists of a fully connected layer $W_{in} \in \mathbb{R}^{Ld \times r}$, a convolutional layer $W_{orth} \in \mathbb{R}^{r \times r \times 1 \times 1}$, and a fully connected layer $W_{out} \in \mathbb{R}^{r \times Ld}$. Here, $r$ is low a rank (default set to 8) to save computational resources. The concatenated features are reduced to a low-dimensional orthogonal space, applying orthogonal constraints and allocating weights:

$$F_{down}^* = W_{in}(F_{con}^*); \quad F_{orth}^* = W_{orth}(F_{down}^*) \qquad (9)$$

where $F_{orth}^* \in \mathbb{R}^{r \times n \times n}$. Next, we compute the orthogonal regularization (Xie et al., 2017) by reshaping $F_{orth}^*$ to $\mathbb{R}^{r \times n^2}$ and using it as a loss term to constrain the extracted pattern, promoting their disentanglement:

$$L_{orth} = \|F_{orth}F_{orth}^T - I\|_F^2 \qquad (10)$$

Finally, we map $F_{orth}^*$ back to the original space and split the concatenated support and query features:

$$F_{up}^* = W_{out}(F_{orth}^*); \quad \{F_l^*\}_{l=1}^L = split(F_{up}^*) \qquad (11)$$

During source-domain training, OSD is trained jointly with the encoder. During target-domain fine-tuning, $W_{in}$ and $W_{out}$ are frozen, and we fine-tune the compact $W_{orth}$.

### 3.2. Cross-Pattern Comparison for Re-Composition

**Cross Comparison.** Based on feature entanglement caused by misaligned matches and the dynamic nature of

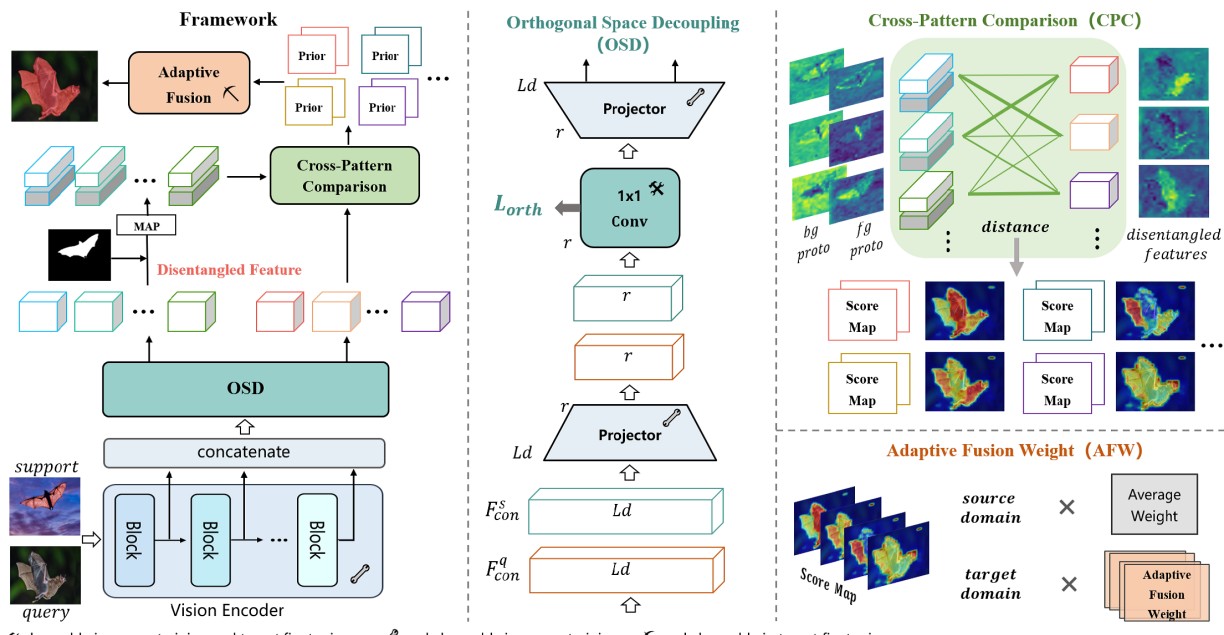

Figure 5: Overview of our method. We extract $L$ support and query features from various blocks. These features are concatenated along the channel dimension and fed into OSD to impose orthogonal constraints for weight allocation and semantic decoupling. Outputs of OSD are then fed into the CPC module, where support and query features are compared crossly, yielding $L \times L$ score maps. In source-domain training, these maps are composed with average weights for efficient pattern learning. In target-domain fine-tuning, the AFW dynamically learns the composition weights for efficient adaption.

ViT, we propose the Cross-Pattern Comparison (CPC) module. After semantic decoupling and weight allocation by the OSD, we use mask average pooling (MAP) (Zhang et al., 2020b) to obtain $L$ sets of foreground prototypes $P_{fg} \in \mathbb{R}^{L \times d \times 1 \times 1}$ and background prototypes $P_{bg} \in \mathbb{R}^{L \times d \times 1 \times 1}$ from support features. These disentangled support prototypes and query features are then input into the CPC module, where they are cross-compared for re-composition.

Specifically, for the query feature sets $F^q \in \mathbb{R}^{L \times d \times n \times n}$ and the support prototypes $[P_{bg}, P_{fg}]$, we compute the distance through cross-pairing to obtain the cross-pattern comparison maps (score maps), denote as $C$:

$$C_{bg/fg} = distance(F^q, P_{bg/fg}); \quad C = concat(C_{bg}, C_{fg}) \quad (12)$$

where $C$ is reshaped to $\mathbb{R}^{L^2 \times 2 \times n \times n}$ (2 means background and foreground), and the distance can be calculated in various ways; we default to using cosine similarity, while also exploring other distance metrics (see Experiment 4.3):

$$distance_{cos} = F^q \cdot P_{bg/fg} / \|F^q\| \|P_{bg/fg}\| \quad (13)$$

**Adaptive Fusion Weight.** For re-composite the obtained $L^2$ sets of cross-pattern comparison maps, during source-domain training, the comparison maps $C$ are composed with average weights for efficient pattern learning; during target-domain fine-tuning, Adaptive Fusion Weight (AFW), which includes background and foreground weight, is introduced

for efficient adaption. The reason for not using AFW during training is that it is a small parameter matrix of size $L^2 \times 2$ (just 288 for ViT-B), where the "2" corresponds to the composition weight for $C_{bg}$ and $C_{fg}$. If trained jointly with the encoder in the source domain, it is prone to overfitting the source data. As a lightweight module, directly introducing it in the target domain allows for flexible adjustment and adaptation based on the target domain, resulting in better performance. The formula is as follows:

$$source: \quad C_{fusion} = \frac{\sum_{l=0}^{L^2} C(l)}{L^2} \quad (14)$$

$$target: \quad C_{fusion} = \frac{W_{AFW} \otimes C}{L^2} \quad (15)$$

where $C(l) \in \mathbb{R}^{2 \times n \times n}$, $\otimes$ indicates the element-wise multiplication. The final prediction $pred$ as describe:

$$pred = argmax(\zeta_l(C_{fusion})) \quad (16)$$

where $\zeta_l(*)$ is a function that bilinearly interpolates $C_{fusion}$ to the spatial size of the input image by expanding along the spatial dimension, i.e., $\zeta_l : \mathbb{R}^{2 \times n \times n} \to \mathbb{R}^{2 \times h \times w}$. Here, $h$ and $w$ are the image's height and width.

**Loss Strategy.** During both source-domain training and target-domain fine-tuning, we employ the standard Binary Cross-Entropy (BCE) loss $L_{BCE}$, with the orthogonal loss $L_{orth}$ from OSD added as a regularization term to promote

| Method | Mark | Backbone | FSS-1000 | | Deepglobe | | ISIC | | Chest X-ray | | Average | |
|---|---|---|---|---|---|---|---|---|---|---|---|---|
| | | | 1-shot | 5-shot | 1-shot | 5-shot | 1-shot | 5-shot | 1-shot | 5-shot | 1-shot | 5-shot |
| PANet (Wang et al., 2019) | ECCV-20 | Res-50 | 69.15 | 71.68 | 36.55 | 45.43 | 25.29 | 33.99 | 57.75 | 69.31 | 47.19 | 55.10 |
| RPMMs (Yang et al., 2020a) | ECCV-20 | Res-50 | 65.12 | 67.06 | 12.99 | 13.47 | 18.02 | 20.04 | 30.11 | 30.82 | 31.56 | 32.85 |
| PFENet (Tian et al., 2020) | TPAMI-20 | Res-50 | 70.87 | 70.52 | 16.88 | 18.01 | 23.50 | 23.83 | 27.22 | 27.57 | 34.62 | 34.98 |
| RePRI (Boudiaf et al., 2021) | CVPR-21 | Res-50 | 70.96 | 74.23 | 25.03 | 27.41 | 23.27 | 26.23 | 65.08 | 65.48 | 46.09 | 48.34 |
| HSNet (Min et al., 2021) | ICCV-21 | Res-50 | 77.53 | 80.99 | 29.65 | 35.08 | 31.20 | 35.10 | 51.88 | 54.36 | 47.57 | 51.38 |
| PATNet (Lei et al., 2022) | ECCV-22 | Res-50 | 78.59 | 81.23 | 37.89 | 42.97 | 41.16 | 53.58 | 66.61 | 70.20 | 56.06 | 61.99 |
| PATNet (Lei et al., 2022) | ECCV-22 | ViT-base | 72.03 | - | 22.37 | - | 44.25 | - | 76.43 | - | 53.77 | - |
| PerSAM (Zhang et al., 2024) | ICLR-24 | ViT-base | 60.92 | 66.53 | 36.08 | 40.65 | 23.27 | 25.33 | 29.95 | 30.05 | 37.56 | 40.64 |
| APM (Tong et al., 2024) | NeurIPS-24 | Res-50 | 79.29 | 81.83 | 40.86 | 44.92 | 41.71 | 51.16 | 78.25 | 82.81 | 60.03 | 65.18 |
| ABCDFSS (Herzog, 2024) | CVPR-24 | Res-50 | 74.60 | 76.20 | 42.60 | 45.70 | 45.70 | 53.30 | 79.80 | 81.40 | 60.67 | 64.97 |
| DRA (Su et al., 2024) | CVPR-24 | Res-50 | 79.05 | 80.40 | 41.29 | **50.12** | 40.77 | 48.87 | 82.35 | 82.31 | 60.86 | 65.42 |
| APSeg (He et al., 2024) | CVPR-24 | ViT-base | 79.71 | 81.90 | 35.94 | 39.98 | 45.43 | 53.98 | **84.10** | 84.50 | 61.30 | 65.09 |
| **SDRC (Ours)** | Ours | ViT-base | **80.31** | **82.55** | **43.15** | 46.83 | **46.57** | **55.02** | 82.86 | **84.79** | **63.22** | **67.30** |

Table 2: Mean-IoU of 1-shot and 5-shot results on the CD-FSS benchmark. The best and second-best results are highlighted in bold and underlined, respectively. **The comparison with domain transfer methods is provided in Appendix C.**

semantic decoupling. Notably, in target-domain finetuning, we do not access the query data. Instead, we treat the support as the query for the calculation of $L_{BCE}$ and $L_{orth}$. We optimize the model using the final loss $L$:

$$L_{BCE} = BCE(pred, y) \tag{17}$$

$$L = L_{BCE} + \lambda L_{orth} \tag{18}$$

where $y$ is the query ground truth in source-domain training and denotes support mask in target-domain fine-tuning, and $\lambda$ is a hyperparameter, with a default value of 0.1, that adjusts the weight of the orthogonal loss $L_{orth}$.

## 4. Experiments

### 4.1. Dataset and Implementation Details

The benchmark proposed by PATNet (Lei et al., 2022) is adopted, following the same data preprocessing procedures as the dataset it employs. PASCAL VOC 2012 (Everingham et al., 2010) with SBD (Hariharan et al., 2011) augmentation serves as the training dataset. FSS-1000 (Li et al., 2020), DeepGlobe (Demir et al., 2018), ISIC2018 (Codella et al., 2019; Tschandl et al., 2018), and Chest X-ray (Candemir et al., 2013; Jaeger et al., 2013) are considered as target domains for evaluation. See the Appendix B for details.

Following previous prototype-based work (Dong & Xing, 2018; Zhang et al., 2020b; Wang et al., 2019), we built a lightweight encoder-only baseline that employs ViT-B (Dosovitskiy et al., 2020) pre-trained on ImageNet (Russakovsky et al., 2015) as the backbone network. The hyperparameter $\lambda$, which adjusts the weight of the orthogonal loss, is set to 0.1, and the rank $r$ of the OSD module is set to 8. For other details, please refer to the appendix.

### 4.2. Comparison with State-of-the-Art Works

In Table 2, we compare our method with existing works, where we achieve a significant improvement under both

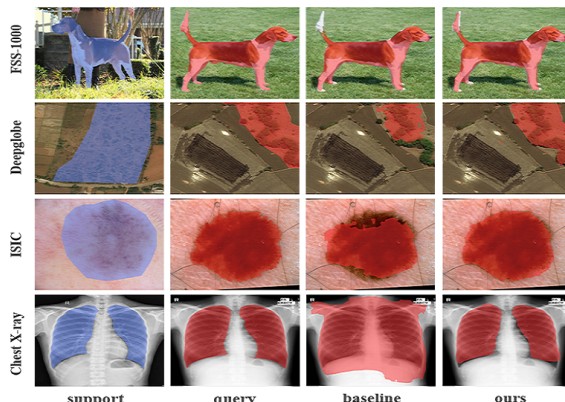

Figure 6: Qualitative results of our model for 1-shot setting. The support labels are highlighted in blue, while the predictions and ground truth of query images are presented in red.

1-shot and 5-shot settings. Specifically, we surpass the performance of the state-of-the-art by 1.92% and 1.88% under 1-shot and 5-shot settings, respectively. Notably, APSeg also utilizes ViT as its backbone; however, its parameter count is significantly larger than ours due to its SAM-based encoder-decoder architecture, while we employ an encoder-only structure. Consequently, our method not only surpasses APSeg in performance but also entails substantially lower computational costs. Furthermore, we showcase the qualitative results of our method in 1-way 1-shot segmentation, as depicted in Figure 6. These results demonstrate the substantial enhancement in generalization ability across significant domain gaps while maintaining a comparable accuracy in the face of similar domain shifts using our method.

### 4.3. Ablation Study

**Impact of each design.** As shown in Table 3, introducing the CPC, which serves as the foundation for the other designs, improved the average mIoU by 9.62% and 9.04%

| CPC | AFW | OSD | FSS-1000 | | Deepglobe | | ISIC | | Chest X-ray | | Average | |
|---|---|---|---|---|---|---|---|---|---|---|---|---|
| | | | 1-shot | 5-shot | 1-shot | 5-shot | 1-shot | 5-shot | 1-shot | 5-shot | 1-shot | 5-shot |
| | | | 77.80 | 80.69 | 33.18 | 37.39 | 36.99 | 41.20 | 51.54 | 55.26 | 49.88 | 53.64 |
| ✓ | | | 79.20 | 81.46 | 41.71 | 43.25 | 42.99 | 47.97 | 74.81 | 78.05 | 59.50 | 62.68 |
| ✓ | ✓ | | 79.22 | 82.02 | 42.59 | 45.22 | 43.11 | 50.73 | 80.36 | 80.36 | 61.32 | 65.22 |
| ✓ | | ✓ | 80.05 | 82.18 | 41.87 | 44.68 | 45.63 | 52.61 | 75.45 | 78.32 | 60.75 | 64.45 |
| ✓ | ✓ | ✓ | **80.31** | **82.55** | **43.15** | **46.83** | **46.5**7 | **55.02** | **82.86** | **84.79** | **63.22** | **67.30** |

Table 3: Detailed ablation study results of our various designs on four target datasets under 1-shot setting and 5-shot setting.

| Metric | baseline | | ours | |
|---|---|---|---|---|
| | 1-shot | 5-shot | 1-shot | 5-shot |
| *Euclidean* | 48.92 | 53.07 | 62.49 | 66.53 |
| *Dot* | 49.18 | 53.03 | 62.75 | 66.58 |
| *EMD* | 50.02 | 53.23 | 63.37 | 67.01 |
| *Cosine* | 49.88 | 53.64 | 63.22 | 67.30 |

Table 4: Impact of the different distance metric for CPC.

| | Encoder (ViT-B) | AFW | OSD (rank=8) | | |
|---|---|---|---|---|---|
| | | | $W_{in}$ | $W_{orth}$ | $W_{out}$ |
| Params(K) | $8.6 \times 10^4$ | 0.288 | 6.144 | 0.064 | 6.144 |
| rank | 64 | 32 | 16 | 8 | 4 | 2 |
| mIoU | 62.61 | 63.43 | 63.25 | 63.22 | 61.73 | 60.39 |

Table 6: Analysis the impact of parameters on performance.

for the 1-shot and 5-shot settings, respectively. The OSD is an explicit global decoupling module, guiding layers to focus on the patterns emphasized by the current layer and promoting disentanglement. Meanwhile, the AFW adjusts the weights of comparison maps for each pattern based on different target domains, resulting in more accurate segmentation predictions. These results demonstrate that each design in our approach significantly enhances performance.

**Impact of different distance metric.** Our method is highly versatile, as we evaluated it using various distance metrics, including Euclidean distance (Snell et al., 2017), cosine similarity (Vinyals et al., 2016), dot product (Chen et al., 2019), and EMD (Zhang et al., 2020a), as shown in Table 4. Regardless of the distance metric, our method consistently outperforms the baseline with significant gains. Notably, under 1-shot setting, EMD is the best distance metric, while cosine similarity performs best under 5-shot setting.

**Effectiveness of cross-comparison.** In Table 5, we report the performance of both the position-wise pattern comparison and cross-pattern comparison, confirming the effectiveness of the cross-layer strategy. Due to the dynamic nature of ViT and intra-class variations, features extracted from different layers may still form correct matches. Therefore, the CPC effectively compares these patterns for re-composition.

| w/o AFW&OSD | 1-shot | 5-shot |
|---|---|---|
| baseline | 49.88 | 53.64 |
| position-wise comparison | 55.14 | 59.39 |
| cross-pattern comparison | 59.50 | 62.68 |

Table 5: Validate the effectiveness of cross-comparison.

**Impact of parameters on performance.** As shown in Table 6, the OSD and the AFW are lightweight modules with minimal parameters. The OSD is trained jointly with the encoder on the source domain, but on the target dataset, only the $W_{orth}$ of OSD is fine-tuned, reducing computational

costs while ensuring effective global decoupling for different domains. The AFW is not involved in source-domain training and is directly adopted during the target stage. We also explored the impact of the OSD's rank value on performance. With rank=8, the performance is slightly lower than 32, but the parameter count is only $\frac{1}{4}$, so we set rank to 8.

**Re-composition strategy for comparison maps.** During source domain training, we composite the comparison maps with average weights. During target domain fine-tuning, the AFW is introduced to adaptively learn the combination weights for efficient adaption. As shown in Table 7, training AFW jointly with the encoder during source-domain training does not achieve better results compared to adapting it directly on the target domain. Training AFW in the source domain can lead to weights being biased toward the source data, so we opt for fine-tuning directly on the target domain.

| AFW | 1-shot | 5-shot |
|---|---|---|
| w/ source-domain training | 61.01 | 64.93 |
| w/o source-domain training | 63.22 | 67.30 |

Table 7: Validation of the training strategy for AFW.

## 4.4. Self-Disentanglement and Re-Composition

**Disentangle Feature by decomposing ViT structure:** To validate our approach, we visualize the features of each layer of ViT-B, as shown in Fig. 7. Since ViT-B has 12 layers, the features are grouped in pairs (with 6 layers per row). The results reveal that the extracted features from different layers capture distinct semantic information, focusing on elements such as the fish tail, body, fins, and outline. The

| OSD | FSS-1000 | | Deepglobe | | ISIC | | ChestX | |
|---|---|---|---|---|---|---|---|---|
| | w/o | w/ | w/o | w/ | w/o | w/ | w/o | w/ |
| support MI | 0.6444 | 0.6059 | 0.8599 | 0.7958 | 0.8738 | 0.7890 | 0.9095 | 0.6506 |
| query MI | 0.6437 | 0.6051 | 0.8593 | 0.7962 | 0.8712 | 0.7829 | 0.9110 | 0.6517 |

Table 8: The MI between features; lower values correspond to lower correlations (better disentanglement).

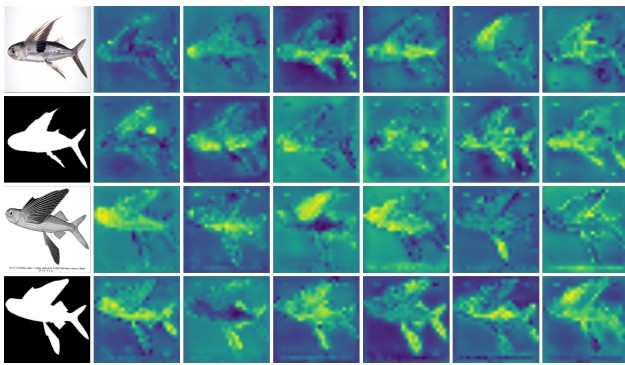

Figure 7: Visualization of features extracted from different layers of ViT demonstrates the feasibility of disentangling the entangled patterns by decomposing the ViT structure.

result demonstrates that ViT inherently has the potential for semantic disentanglement and supports the feasibility of our insight: decompose the entangled semantic patterns through a structural decomposition of the ViT output.

**Further semantic disentanglement by OSD:** Mutual information reflects the correlation between features. Lower mutual information indicates weaker correlations, meaning the features are more semantically independent. Therefore, we measure the average mutual information (MI) separately between the support features and between the query features to verify that OSD promotes further semantic disentanglement. As shown in Table 8, after applying OSD, the mutual information between both support features and between query features decreases, verifying OSD's decoupling.

**Cross-Pattern Comparison for re-composition:** In Fig. 8, we visualize the results of cross-comparison (some samples) and re-composition. Through cross-comparison, the model focuses on different regions of the segmented object. These comparison maps are then re-composed, allowing the model to accurately identify the complete segmentation region.

**Re-composition by Adaptive Fusion Weight:** We visualize the AFW as heatmaps, where brighter colors indicate higher re-composition weights. The result in Fig.9 highlights that: 1) AFW learns different re-composition weights for each domain; 2) The highest weight learned by AFW is not necessarily at the diagonal of the matrix, suggesting that cross-matching and re-composition are more effective than position-wise matching. Additionally, we observe an interesting phenomenon: without any constraints imposed on AFW, the learned re-composition weights for foreground and background tend to be mutually exclusive, a trend particularly noticeable in the Deepglobe and ISIC datasets.

## 5. Analysis of Performance and Efficiency

**Orthogonal Loss Weight** As shown in Table 9, we validated the impact of the weight of the orthogonal loss on

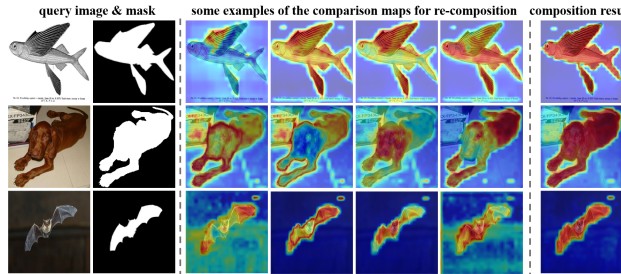

Figure 8: The heatmaps of some examples of cross-comparison maps and the results after re-composition.

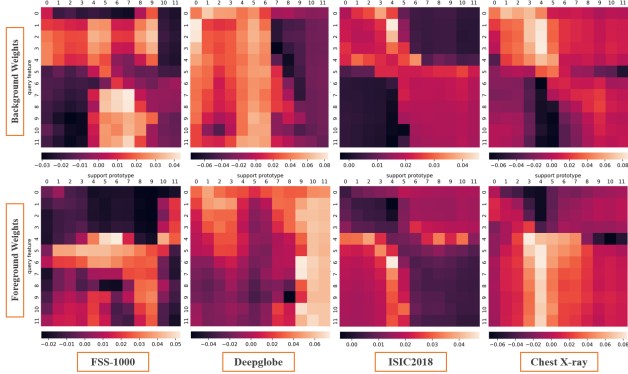

Figure 9: Visualization of AFW on four target datasets, which includes background and foreground weight. A brighter color indicates a higher re-composition weight.

performance. The results indicate that the optimal choice of the weight falls within a wide interval, which means the tuning of this hyper-parameter is not difficult. Additionally, we used the same orthogonal loss weight in the Swin Transformer architecture as we did in the ViT architecture (Table 17). The performance indicates that our method design is not sensitive to this weight.

| Orth. Loss Weight | 0.01 | 0.05 | 0.1 | 0.2 | 0.5 |
|---|---|---|---|---|---|
| 1-shot Avg mIoU | 62.59 | 63.01 | 63.22 | 63.18 | 62.87 |

Table 9: Impact of orthogonal loss weight on performance.

**Impact of Background Prototypes's Number** Most current prototype-based methods (e.g., PANet, SSP) utilize a single background prototype to model background patterns, and have demonstrated good performance in both recent works and our experiments. Indeed, it is preferable to consider different background classes for different images. Therefore, as shown in Table 10, we further introduce clustering to obtain multiple background prototypes (Yang et al., 2020b; Li et al., 2021a). By comparing the single-background prototype to multi-background prototypes, we observe a slight performance improvement. However, the gains are not substantial enough to justify the additional computational overhead of clustering.

|  | FSS1000 | Deepglobe | ISIC | ChestX | Mean |
|---|---|---|---|---|---|
| Single BG prototype | 80.31 | 43.15 | 46.57 | 82.86 | 63.22 |
| Multi BG prototype | 80.96 | 43.53 | 46.78 | 83.09 | 63.59 |

Table 10: Comparison between single and multiple background prototypes under 1-shot setting.

**Computational Efficiency** As shown in Table 11, we compared our method with PATNet (Lei et al., 2022), HSNet (Min et al., 2021), and SSP (Fan et al., 2022). Our approach exhibits greater computational efficiency than the other methods, as it does not require additional networks and instead leverages the inherent structure of the ViT for feature separation.

|  | PATNet | HSNet | SSP | Ours |
|---|---|---|---|---|
| FLOPs (G) | 22.63 | 20.11 | 18.97 | **18.86** |

Table 11: Analysis of computational efficiency.

## 6. Theoretical Analysis of the Effectiveness of ViT Disentanglement

In cross-domain few-shot segmentation tasks, models need to transfer knowledge from the **source domain** $\mathcal{S}$ with abundant annotations to the **target domain** $\mathcal{T}$ with limited data. Let $\mathcal{H}$ represent the hypothesis space of the segmentation model. The upper bound of the generalization error for target domain risk $\epsilon_{\mathcal{T}}(h)$ is defined as:

$$\epsilon_{\mathcal{T}}(h) \leq \epsilon_{\mathcal{S}}(h) + d_{\mathcal{H}}(\mathcal{S}, \mathcal{T}) + \lambda, \tag{19}$$

where $h$ denotes features extracted by the encoder, $\epsilon_{\mathcal{S}}(h)$ is the source domain risk, $d_{\mathcal{H}}(\mathcal{S}, \mathcal{T})$ represents the $\mathcal{H}$-divergence (domain gap) between the source and target domains, and $\lambda$ is the irreducible ideal joint risk.

Our approach reduces $\epsilon_{\mathcal{T}}(h)$ through two mechanisms:

*1) Adaptive Fusion Weights (AFW):* adaptively assigns higher weights to semantically appropriate matches, leading to better alignment (as confirmed by the experiments on the source domain in Answer 3), thereby optimizing source domain output and reducing $\epsilon_{\mathcal{S}}(h)$.

*2) Domain-Invariant Component Isolation:* minimizes $d_{\mathcal{H}}(\mathcal{S}, \mathcal{T})$ by isolating domain-invariant patterns (e.g., object parts) via:

$$d_{\mathcal{H}}(\mathcal{S}, \mathcal{T}) \approx \sum_{i=1}^{L} \sum_{j=1}^{L} w_{ij} d_{\mathcal{H}}^{(ij)}(\mathcal{S}, \mathcal{T}) \leq \sum_{i=1}^{L} \sum_{j=1}^{L} d_{\mathcal{H}}^{(ij)}(\mathcal{S}, \mathcal{T}), \tag{20}$$

where $d_{\mathcal{H}}^{(ij)}$ denotes the inter-layer domain discrepancy. By leveraging the self-disentangling property and the orthogonal constraints from the OSD module, inappropriate matches are learned to have small $w_{ij}$, reducing the inter-layer mutual information $\mathbb{I}[\mathbf{h}_i \mathbf{h}_j^T]$, thereby tightening the boundary.

## 7. Related Work

**Cross-Domain Few-Shot Segmentation (CD-FSS)** CD-FSS has received increasing attention recently. PAT-Net (Lei et al., 2022) establishes a CD-FSS benchmark and proposes feature transformation layers to map domain-specific features into domain-agnostic ones for fast adaption. APSeg (He et al., 2024), based on SAM (Kirillov et al., 2023), introduces a novel auto-prompt network for guiding features in cross-domain segmentation. DRA (Su et al., 2024) adopts a compact adapter to align diverse target domain features with the source domain, while ABCDFSS (Herzog, 2024) introduces tiny adaptors that learn to refine features at test-time only. APM (Tong et al., 2024) proposes a lightweight frequency masker to achieve feature enhancement. These methods focus on optimizing encoders' final features to obtain a better representation. In contrast, our approach focuses on decomposing ViT's final features based on the natural decomposition of ViT's structure, and re-composing them for better comparison.

**Feature Disentanglement Learning (FDL)** FDL aims to learn an interpretable representation for image variants, which has long been a popular solution for addressing domain shifts. InfoGAN (Chen et al., 2016) maximizes mutual information to learn disentangled representations in an unsupervised manner. Disentangled-VAE (Li et al., 2021b) excavate category-distilling information from visual and semantic features for generalized zero-shot learning. DFR (Cheng et al., 2023) separates discriminative features from class-irrelevant components. However, these methods require additional complex VAE-Discriminator networks, resulting in significant computational overhead. In contrast, our approach does not introduce any extra branch networks. Instead, it leverages feature space consistency across different layers of the ViT to extract distinct patterns, decomposing the entangled semantic patterns through a structural decomposition of the ViT output from a novel perspective.

## 8. Conclusion

In this paper, we analyze and interpret the feature entanglement problem from a novel aspect of the natural decomposition of ViT. Based on it, we propose self-disentanglement and re-composition for CD-FSS. Experiments show our effectiveness and achieve a new state-of-the-art in CD-FSS.

## Acknowledgements

This work is supported by the National Key Research and Development Program of China under grant 2024YFC3307900; the National Natural Science Foundation of China under grants 62206102, 62436003, 62376103 and 62302184; the National Natural Science Foundation of China under grants 62402015; the Postdocotoral Fellowship Program of CPSF under grants GZB20230024;

the China Postdoctoral Science Foundation under grant 2024M750100; Major Science and Technology Project of Hubei Province under grant 2024BAA008; Hubei Science and Technology Talent Service Project under grant 2024DJC078; and Ant Group through CCF-Ant Research Fund. The computation is completed in the HPC Platform of Huazhong University of Science and Technology.

## Impact Statement

Given current interpretations of ViT structures, we find a natural decomposition exists in features extracted by ViT, which inspires us to propose the concept of self-disentanglement and re-composition of ViT features. Our research can also be applied in other fields such as transfer learning and domain adaptation. What's more, our method also provides an interpretable perspective and theoretical foundation for the feature disentanglement from the perspective of ViT's self-decomposition. Future research will aim to broaden our evaluations to encompass a wider range of target domains, enhancing our understanding of their performance in various real-world scenarios.

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

# Appendix for Self-Disentanglement and Re-Composition for Cross-Domain Few-Shot Segmentation

## A. Centered Kernel Alignment (CKA)

Centered Kernel Alignment (CKA) (Kornblith et al., 2019) is a widely used metric for measuring the similarity between two data representations (Zou et al., 2022; 2024b; Liu et al., 2025b). It normalizes the Hilbert-Schmidt Independence Criterion (HSIC) to mitigate scale differences and ensure a more stable similarity measure.

HSIC quantifies dependence between two sets of features in a reproducing kernel Hilbert space (RKHS). For centered feature matrices $\mathbf{X}$ and $\mathbf{Y}$, Equation (1) gives:

$$\frac{1}{(n-1)^2}\mathrm{tr}(\mathbf{X}\mathbf{X}^\top\mathbf{Y}\mathbf{Y}^\top) = |\mathrm{cov}(\mathbf{X}^\top, \mathbf{Y}^\top)|F^2. \quad (21)$$

HSIC extends this to kernel-based methods, where $\mathbf{K}ij = k(x_i, x_j)$ and $\mathbf{L}_{ij} = l(y_i, y_j)$ define kernel matrices. The empirical HSIC estimator is:

$$\mathrm{HSIC}(\mathbf{K}, \mathbf{L}) = \frac{1}{(n-1)^2}\mathrm{tr}(\mathbf{K}\mathbf{H}\mathbf{L}\mathbf{H}), \quad (22)$$

where $\mathbf{H}$ is the centering matrix:

$$\mathbf{H} = \mathbf{I}_n - \frac{1}{n}\mathbf{1}\mathbf{1}^\top. \quad (23)$$

Here, $\mathbf{I}_n$ is the identity matrix, and $\mathbf{1}$ is a vector of ones. HSIC effectively quantifies statistical dependence and converges to its population estimate at a rate of $1/\sqrt{n}$.

To address HSIC's sensitivity to scale, CKA introduces normalization. The CKA metric between two kernel matrices $\mathbf{K}$ and $\mathbf{L}$ is:

$$\mathrm{CKA}(\mathbf{K}, \mathbf{L}) = \frac{\mathrm{HSIC}(\mathbf{K}, \mathbf{L})}{\sqrt{\mathrm{HSIC}(\mathbf{K}, \mathbf{K}) \cdot \mathrm{HSIC}(\mathbf{L}, \mathbf{L})}}. \quad (24)$$

The numerator measures the similarity between the two kernels, while the denominator normalizes it using self-similarities within each representation. This normalization ensures CKA's invariance to isotropic scaling, making it a robust similarity measure for feature representations.

## B. More Details about Datasets

We adopt the benchmark established by PATNet (Lei et al., 2022). Fig. 10 illustrates segmentation examples for four target datasets. Further details are as follows:

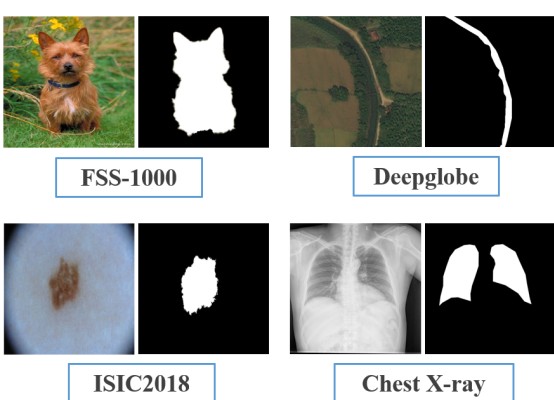

**FSS-1000**          **Deepglobe**

**ISIC2018**          **Chest X-ray**

Figure 10: Examples of images and their corresponding ground truth masks from four target domain datasets.

**PASCAL**-$5^i$ (Shaban et al., 2017) is an extended version of PASCAL VOC 2012 (Everingham et al., 2010), incorporating additional annotation details from the SDS dataset (Hariharan et al., 2011). We use PASCAL as the source-domain dataset for model training and then evaluate the performance on four target-domain datasets.

**FSS-1000** (Li et al., 2020) is a natural image dataset, encompassing 1,000 distinct categories, each represented by 10 samples. In this study, we adhere to the official dataset split for semantic segmentation and report our results on the specified test set, which comprises 240 classes and a total of 2,400 images. FSS-1000 is employed as the target domain for performance evaluation.

**Deepglobe** (Demir et al., 2018) is a dataset comprising satellite imagery with dense pixel-level annotations across seven categories: urban, agriculture, rangeland, forest, water, barren, and unknown. Since ground-truth labels are only available for the training set, we utilize the official training dataset, which includes 803 images, for evaluation. We adopt Deepglobe as the target domain for evaluation and follow the same processing methodology as PATNet.

**ISIC2018** (Codella et al., 2019; Tschandl et al., 2018) is a skin cancer screening dataset, consisting of lesion images where each image contains a single primary lesion. The dataset is processed and utilized following the standards established by PATNet. We consider ISIC2018 as the target domain for evaluation.

**Chest X-ray** (Candemir et al., 2013; Jaeger et al., 2013)

is a dataset for Tuberculosis screening, consisting of 566 high-resolution images (4020 × 4892 pixels). These images are drawn from 58 Tuberculosis cases and 80 normal cases. To handle the large image dimensions, we resize them to 1024 × 1024 pixels.

## C. Comparison with Domain Transfer Methods

We compare our method with traditional disentanglement representation methods and multi-layer fusion approaches to validate its effectiveness. For a fair comparison, all methods are implemented on the same baseline and evaluated under the 1-shot setting on the CD-FSS benchmark.

**Disentanglement representation methods**   Disentanglement representation methods has long been a popular solution for addressing domain shifts. InfoGAN (Chen et al., 2016) maximizes mutual information to learn disentangled representations in an unsupervised manner. Disentangled-VAE (Li et al., 2021b) excavate category-distilling information from visual and semantic features for generalized zero-shot learning. DFR (Cheng et al., 2023) separates discriminative features from class-irrelevant components. However, these methods require additional complex VAE-Discriminator networks, resulting in significant computational overhead. Moreover, under the few-shot setting, these methods struggle to leverage limited data to learn a suitable latent space for disentanglement. In contrast, our approach does not introduce any extra branch networks. Effectively addresses few-shot scenarios while maintaining low computational overhead. As shown in Table 12 our method outperforms existing disentanglement-based approaches on the CD-FSS task.

|  | FSS | Deepglobe | ISIC | Chest | Average |
|---|---|---|---|---|---|
| baseline | 77.80 | 33.18 | 36.99 | 51.54 | 49.88 |
| InfoGAN | 78.73 | 35.62 | 38.19 | 65.45 | 54.50 |
| Disentangled-VAE | 78.67 | 36.02 | 37.79 | 66.38 | 54.72 |
| DFR | 79.18 | 39.21 | 40.62 | 72.85 | 57.97 |
| SDRC (Ours) | **80.31** | **43.15** | **46.57** | **82.86** | **63.22** |

Table 12: Compare our method to previous disentanglement-based methods under 1-shot setting.

**Feature fusion methods**   FPN (Lin et al., 2017) proposes an in-network feature pyramid architecture that enhances semantic richness across all scales by combining high- and low-resolution features. MEP3P (Zhu et al., 2024) enhanced the original visual features input into MLLMs with image depth features and pseudo-3D positions. MMFuser (Cao et al., 2024) integrated features from multiple layers, enriching the visual inputs for MLLMs by capturing multi-level representations from the vision encoder. These methods enhance the encoder's output representation by utilizing multi-layer information. They aims to increase the feature's

robustness and discrimination by aggregating information from multiple layers. In contrast, our approach decouples the encoder's output into independent semantic representations, which improves transferability in cross-domain settings. As shown in Table 13 our method outperforms existing fusion-based approaches on the CD-FSS task (for multimodal methods, we apply this approach solely to improve the vision encoder).

|  | FSS | Deepglobe | ISIC | Chest | Average |
|---|---|---|---|---|---|
| baseline | 77.80 | 33.18 | 36.99 | 51.54 | 49.88 |
| FPN | 78.73 | 37.51 | 37.64 | 69.59 | 55.87 |
| MEP3P | 79.96 | 42.92 | 40.43 | 73.87 | 59.30 |
| MMFuser | 80.29 | 38.65 | 42.01 | 75.33 | 59.07 |
| SDRC (Ours) | **80.31** | **43.15** | **46.57** | **82.86** | **63.22** |

Table 13: Compare our method to previous fusion-based methods under 1-shot setting.

**Slot attention-based methods**   Our approach differs from slot attention-based methods (Locatello et al., 2020; Seitzer et al., 2022) in two fundamental aspects: 1) Slot attention primarily disentangles distinct objects through object-centric representation optimization, exhibiting coarser granularity, whereas our method focuses on disentangling different patterns within the same object at a finer granularity level. 2) While slot attention employs additional iterative attention modules (external networks) for disentanglement, we leverage the inherent property of ViT layers that naturally attend to distinct spatial regions, augmented with orthogonality constraints to reinforce semantic separation, without requiring additional networks. To validate the effectiveness of our method, we conduct comparative evaluations against two representative slot attention-based disentanglement approaches, Slot-Attention (Locatello et al., 2020) and DINOSAUR (Seitzer et al., 2022), as shown in Table 14.

|  | FSS1000 | DeepGlobe | ISIC | ChestX | Mean |
|---|---|---|---|---|---|
| Baseline | 77.80 | 33.18 | 36.99 | 51.54 | 49.88 |
| Slot-Attention | 79.05 | 37.83 | 41.22 | 69.59 | 56.92 |
| DINOSAUR | 79.62 | 38.57 | 40.89 | 72.33 | 57.85 |
| SDRC (Ours) | **80.31** | **43.15** | **46.57** | **82.86** | **63.22** |

Table 14: Comparison with slot attention-based methods under 1-shot setting.

## D. Comparison with Methods under Special Setting

Existing CD-FSS methods adopt a batch size of 1 during testing to prevent unfair advantages from incorporating information from other samples. However, IFA (Nie et al., 2024) uses a batch size of 96 during testing, which causes it to calculate foreground and background prototypes by aggregating all samples in a batch. For a fair comparison, we conduct comparisons with IFA using a batch size of 96.

| Method | FSS-1000 | | Deepglobe | | ISIC | | Chest X-ray | |
|---|---|---|---|---|---|---|---|---|
| | 1-shot | 5-shot | 1-shot | 5-shot | 1-shot | 5-shot | 1-shot | 5-shot |
| IFA | 80.1 | 82.4 | 50.6 | 58.8 | 66.3 | 69.8 | 74.0 | 74.6 |
| **Ours** | **83.1** | **85.7** | **51.1** | **59.4** | **69.7** | **72.5** | **84.1** | **87.2** |

Table 15: Comparison with IFA under its specific testing setting, which uses a batch size of 96.

## E. Effectiveness in Swin Transformers

| | Stage 1 | Stage 2 | Stage 3 | Stage 4 |
|---|---|---|---|---|
| Shape | H/4×W/4×C | H/8×W/8×2C | H/16×W/16×4C | H/32×W/32×8C |
| Layer Num | 2 | 2 | 18 | 2 |

Table 16: Configuration of Swin-B transformer architecture.

As shown in Table 17, we further validated the effectiveness of our method on Swin Transformer (Liu et al., 2021). Since the layers in Swin Transformer do not reside in the same feature space, two additional steps are required: (1) For features from stage 2 to stage 4, we upsample them spatially to $H/4 \times W/4$; (2) For features from stage 1 to stage 3, we add three mapping linear layers $(C, 8C)$, $(2C, 8C)$, and $(4C, 8C)$ to map them to the same feature space as that of stage 4. These mapping layers are trained alongside the model during the source domain training phase. The performance results demonstrate that our method is well-suited to Swin Transformer, and that Swin Transformer shows a significant improvement in performance compared to ViT.

| | FSS1000 | Deepglobe | ISIC | ChestX | Mean |
|---|---|---|---|---|---|
| ViT-B | 77.80 | 33.18 | 36.99 | 51.54 | 49.88 |
| Swin-B | 79.85 | 37.24 | 39.90 | 66.73 | 55.93 |
| Swin-B + Ours | **81.02** | **46.63** | **49.19** | **83.85** | **65.17** |

Table 17: Performance under 1-shot setting with Swin Transformer architecture.

## F. Applications in Other Settings

### 1) Benefit for few-shot segmentation task

We measure the FSS performance of our methods on Pascal, which consists of 20 classes and is set to a 4-fold configuration in the FSS setup. This means training is conducted on 5 classes, while testing is performed on 15 classes that were not seen during the training phase. The experimental results show that our method can also effectively improve the performance of general FSS tasks.

### 2) Benefit for domain generalization task

Under the domain generalization setting, our method trained on Pascal and tested on FSS1000 (with removed support sets and finetuning stage) demonstrates that feature disentanglement enhances model generalizability, yielding concomitant

| 1 shot | Fold0 | Fold1 | Fold2 | Fold3 | Mean |
|---|---|---|---|---|---|
| Baseline | 61.5 | 68.2 | 66.7 | 52.5 | 62.2 |
| Ours | **63.1** | **70.3** | **67.8** | **55.4** | **64.2** |

Table 18: Performance on Pascal under FSS setting.

benefits for domain generalization.

| | Baseline | Ours |
|---|---|---|
| FSS | 72.6 | **75.3** |

Table 19: Performance under domain generalization setting.

## G. Connection to More fields

As deep learning advances and finds broader application across AI, ensuring model reliability—including interpretability (Liu et al., 2025a; 2024b), causal reasoning (Liu et al., 2023b; 2024a; 2023a), and robustness (Liu et al., 2023c; 2022; Deng et al., 2023)—has become increasingly important. We recognize this and view it as a promising direction for future work.

## H. More Visualization Results about Self-Disentanglement and Re-Composition

Based on the feature space consistency, we find a natural decomposition in ViT's output, which inspires us to propose the concept of self-disentanglement and re-composition of ViT features for the CD-FSS task, which disentangles features without the need for additional branch networks. In the main text, we presented some visualization examples of self-disentanglement and re-composition. Here, we provide additional examples to further validate our insights.

**Disentangle Feature by decomposing ViT structure:** To validate our approach, we visualize the features of each layer of ViT-B, as shown in Figure 11. The outputs of all 12 layers of ViT-B show that features extracted at different layers exhibit distinct semantic tendencies, focusing on different regions of the segmented object (e.g. the outline, head, wings, body, and tail of a bird). The result demonstrates that ViT inherently has the potential for semantic disentanglement and supports the feasibility of our insight: decompose the entangled semantic patterns through a structural decomposition of the ViT output.

**Cross-Pattern Comparison for re-composition:** Due to the dynamic nature of ViT, the patterns captured by the different layer may be semantically similar (as demonstrated through experiments in the main text). Therefore, we adopt the Cross-Pattern Comparison (CPC) module,

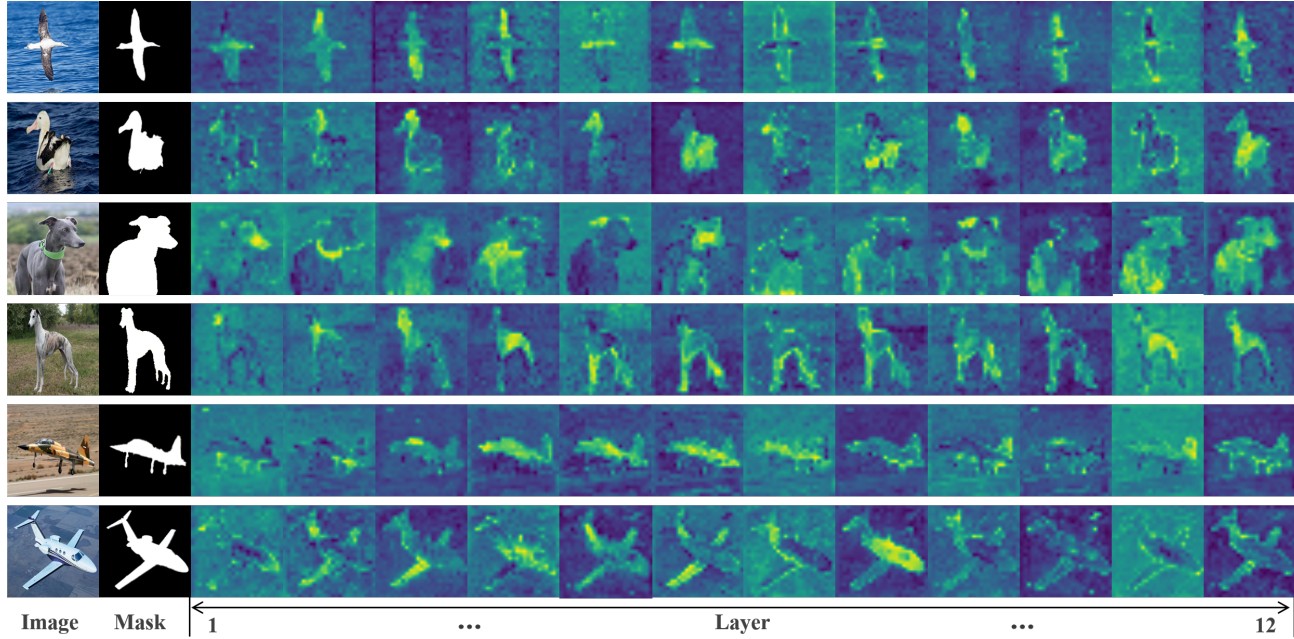

Figure 11: Visualization of features extracted from different layers of ViT demonstrates the feasibility of disentangling the entangled patterns by decomposing the ViT structure.

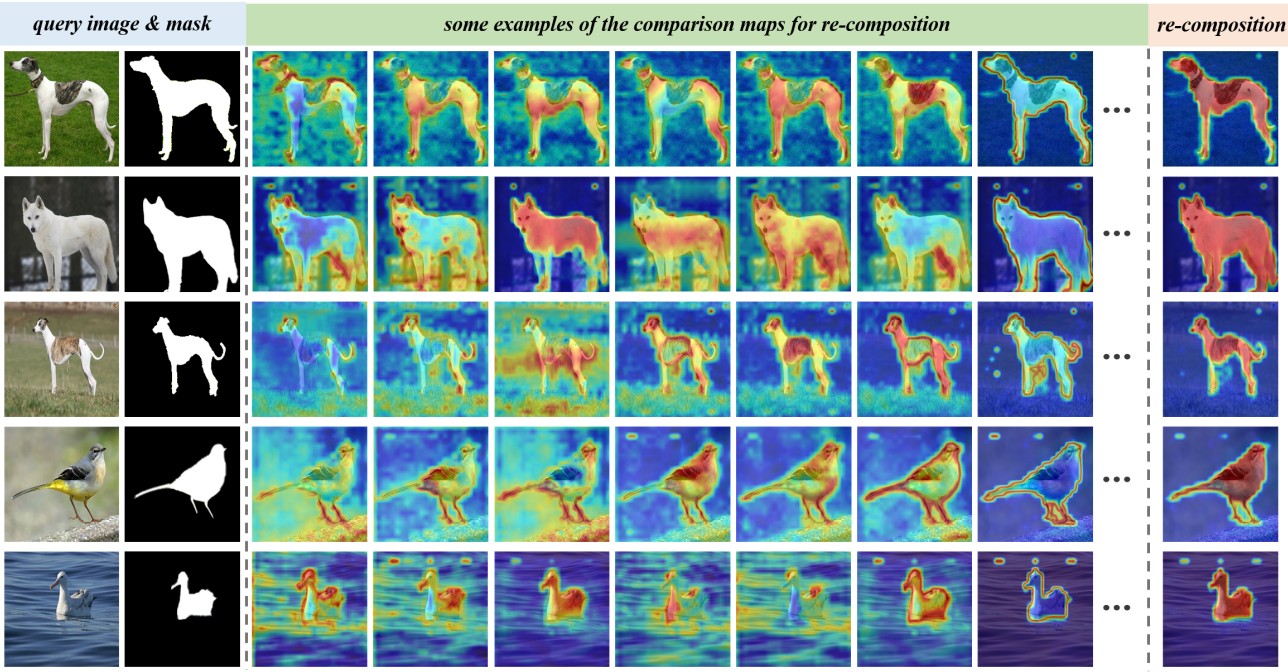

Figure 12: Visualization of the heatmaps showing some examples of cross comparison maps and the results after re-composition.

where the disentangled patterns are cross-compared to facilitate effective re-composition. Here, we present additional cross-comparison (some samples) visualizations and re-composition result visualizations, as shown in Figure 12. Through cross-comparison, the model focuses on different regions of the segmented object. These comparison maps are then re-composed, allowing the model to accurately identify the complete segmentation region.

