# OpenReview forum: "Self-Disentanglement and Re-Composition for Cross-Domain Few-Shot Segmentation"
_ICML.cc/2025/Conference — ICML 2025 poster_

### Official Review · Reviewer_4QAQ · 2025-03-11

**Overall Recommendation:** 3

**Summary:**

The paper finds that previous approaches have an entanglement problem, which tends to bind source domain patterns together, making each one difficult to transfer. They analyzed and explained the feature entanglement problem from a new perspective of the natural decomposition of ViT. On this basis, self-disentanglement and re-composition of CD-FSS are proposed.

### update after rebuttal
I would like to thank the authors for taking the time to address my questions and for providing the additional numerical experiments. I am satisfied with the authors' rebuttal. So I will keep my positive rating.

**Claims And Evidence:**

Yes

**Essential References Not Discussed:**

None

**Experimental Designs Or Analyses:**

Yes

**Methods And Evaluation Criteria:**

Yes

**Other Comments Or Suggestions:**

I suggest the authors test the performance of the model in the FSS setting as well. See if the model can achieve good results in different Settings.

**Other Strengths And Weaknesses:**

Strengths:
The motivation is clear, and the approach is sensible, straightforward, and highly applicable. The paper is easy to understand, the results are easy to reproduce, and a large number of ablation experiments have been done to prove the effectiveness of the results. Paper is well organized. The proposed approach for self-disentanglement and re-composition of CD-FSS from the perspective of natural decomposition of ViT is novel.


Weaknesses：
1. It seems that the network decoupled the CD-FSS task into two tasks: Cross-Domain and FSS, and the model focused more on the FSS part, while there was not much special design for Cross-Domain.
2. I am curious about the portability of the proposed method, the authors did not test the feasibility of the proposed method on other existing methods.
3. Failure cases are not provided, which makes a complete discussion of the approach lacking.
4. The article uses the background prototype in CPC, I am doubtful about the role of background prototype. Because unlike the foreground, the background of the support image and the background of the query image are not constrained to the same class.
5. The $W_{in}$ in Equation 11 should be $W_{out}$.
6. It doesn't describe how to use the network under 5-shot setting.

**Questions For Authors:**

See Weaknesses.

**Relation To Broader Scientific Literature:**

They analyzed and explained the feature entanglement problem from a new perspective of the natural decomposition of ViT. On this basis, self-disentanglement and re-composition of CD-FSS are proposed.

**Theoretical Claims:**

In fact, the authors present no new theories.

---

> ### Author Rebuttal · Authors · 2025-04-01
>
> ## 1. Design for the cross-domain part
>
> Our method design also highlights the cross-domain part of CDFSS in the following aspects.
>
> (1) Intuitively, for the cross-domain transferability, the overall semantics, like a whole bat in Fig.1, is much harder to transfer than the separated parts of it, like wings, claws. Based on this intuition, our method disentangles the overall semantics into small and transferable parts, and recomposes them on the target domain through efficient tuning of AFW.
>
> (2) Quantitatively, we validated in Fig.4 and Tab.1 that it is the mismatch between layers that harms the transferability between source and target domains by the domain similarity via CKA metrics. Therefore, our design addresses the mismatch problem and encourages the model to focus on more meaningful matches (typically with higher CKA value), and is verified in Fig.9 to show that our model can indeed highlight these matches, improving the cross-domain transferability.
>
> ## 2. Portability of our methods
>
> Our method has excellent adaptability to other approaches. To verify this, we combined our method with APM and PATNet. Additionally, in response to reviewer HFkz and PmpS, we applied our method to Swin Transformer, demonstrating its suitability for different transformer architectures.
>
> | 1-shot（ViT-B） |  FSS1000  | Deepglobe |   ISIC    |  ChestX   | Mean      |
> | :-------------: | :-------: | :-------: | :-------: | :-------: | --------- |
> |       APM       |   79.84   |   39.78   |   50.37   |   79.29   | 62.32     |
> |   APM + Ours    | **80.73** | **43.67** | **52.82** | **83.01** | **65.06** |
> |     PATNet      |   72.03   |   22.37   |   44.25   |   76.43   | 53.77     |
> |  PATNet + Ours  | **80.50** | **43.18** | **46.62** | **82.49** | **63.20** |
>
> ## 3. Failure cases
>
> Here https://anonymous.4open.science/r/Self-Disentangle-Rebuttal-for-ICML25-706E/failure_cases.PNG, we analyze some failure cases. We found that in DeepGlobe, areas with large contiguous regions tend to experience incomplete segmentation, particularly at the edges, where the segmentation granularity may not be fine enough. The main reasons are twofold: 1) DeepGlobe consists of remote sensing images, which are typically high-resolution, yet for computational efficiency, we standardize the images to 400x400; 2) DeepGlobe demands high local feature recognition capability from the model. ViT-based methods, due to their attention mechanism, are strong in global modeling but relatively weaker in recognizing local features compared to models like ResNet, which have inherent local priors. Therefore, we believe that future research could focus on enhancing ViT's ability to recognize local features in cross-domain scenarios.
>
> ## 4. Background prototype
>
> #### (1) Role of the background prototype
>
> The final result of segmentation is based on the probability map derived from both the support-set foreground prototype and the support-set background prototype compared with the query feature, rather than relying solely on the background prototype. Thus, the background prototype is complementary to the foreground prototype.
>
> #### (2) Support-set image and query-set image in the same background class
>
> Although the background of support and query image are not in the same class, most current prototype-based methods（PANet, SSP) utilize a single background prototype to model the background patterns, and have been verified to have good performance in both current works and our paper.
>
> Indeed, it is better to consider different background classes for different images. Therefore, here we further utilize clustering to obtain multiple background prototypes. By comparing the single-background prototype versus multi-background prototypes, we observed a slight performance improvement. However, the gains were not substantial enough to justify the computational overhead of clustering. Although this is not the main focus of our paper, we still appreciate your suggestion.
>
> |       1-shot        | FSS1000 | Deepglobe | ISIC  | ChestX | Mean  |
> | :-----------------: | :-----: | :-------: | :---: | :----: | ----- |
> | single BG prototype |  80.31  |   43.15   | 46.57 | 82.86  | 63.22 |
> | multi BG prototype  |  80.96  |   43.53   | 46.78 | 83.09  | 63.59 |
>
> ## 5. Utilization under 5-shot setting
>
> For the 5-shot setting, the methodology is consistent with the 1-shot approach, with the following differences: 1) During fine-tuning, each task has 5 support images available for learning; 2) Five support foreground prototypes are calculated for the same class and then aggregated by averaging.
>
> ## 6. Minor writing error
>
> Thank you very much for your careful correction. Equation 11 should indeed be $W_{out}$ instead of $W_{in}$​. We will take extra care to correct this in the final version to prevent any minor writing errors.
>
> ## 7. FSS performance
> In our response to reviewer PmpS's third point, we validated the effectiveness of our method in the FSS setting. Thank you for your suggestion.

---

> > ### Comment · Reviewer_4QAQ · 2025-04-02
> >
> > I would like to thank the authors for taking the time to address my questions and for providing the additional numerical experiments. I am satisfied with the authors' rebuttal.

---

> > > ### Author Response · Authors · 2025-04-03
> > >
> > > Thank you again for your time in reviewing our paper. We also sincerely appreciate your acknowledgment of our rebuttal responses. As you indicated no remaining questions and comments, we would be truly grateful if you could reconsider to raise your evaluation score. Your generous reconsideration would mean a lot to our research team.

---

### Official Review · Reviewer_HFkz · 2025-03-14

**Overall Recommendation:** 3

**Summary:**

This paper addresses the challenge of feature entanglement in Cross-Domain Few-Shot Segmentation by leveraging the inherent structure of Vision Transformers. The authors identify that current methods often suffer from entangled semantic patterns, which hinder the transferability of features across domains. To tackle this, the paper proposes a framework that disentangles semantic patterns by analyzing ViT natural decomposition and re-composing these disentangled features to improve generalization and adaptation. The approach introduces mechanisms to reduce feature correlation, learns meaningful comparisons across layers, and dynamically adjusts feature weights during fine-tuning for better cross-domain segmentation. The method achieves state-of-the-art results across multiple benchmarks. Extensive experiments demonstrate the effectiveness of the proposed approach.

**Claims And Evidence:**

1. Mutual information analysis shows reduced correlations between disentangled features after applying OSD.
2. Quantitative results on four benchmarks demonstrate significant improvements, supported by qualitative visualizations.
3. Ablation studies show consistent performance gains when CPC and AFW are added.

**Essential References Not Discussed:**

None

**Experimental Designs Or Analyses:**

1. The experimental design is robust, with extensive ablation studies to isolate the contributions of each module.
2. The use of CKA similarity and mutual information to validate disentanglement is effective and aligns with the claims.

**Methods And Evaluation Criteria:**

1. The proposed methods are well-motivated.
2. The experimental setup is comprehensive, with comparisons to strong baselines and state-of-the-art methods.

**Other Comments Or Suggestions:**

None

**Other Strengths And Weaknesses:**

Strengths:
1. The idea of leveraging ViT’s structural decomposition for disentanglement is novel and insightful.
2. The proposed method achieves state-of-the-art performance with significant improvements across benchmarks.

Weakness:
1. The orthogonal loss weight is manually tuned, which may limit adaptability.

**Questions For Authors:**

Can the proposed framework be applied to other ViT architectures (e.g., Swin Transformer)? If so, what modifications are required?

**Relation To Broader Scientific Literature:**

The proposed approach adds a novel perspective by leveraging ViT’s inherent structure for disentanglement and re-composition.

**Theoretical Claims:**

The theoretical analysis of feature entanglement is convincing.

---

> ### Author Rebuttal · Authors · 2025-04-01
>
> ## 1. Orthogonal loss weight
>
> We validated the impact of the weight of the orthogonal loss on performance. The results indicate that the optimal choice of the weight is in a wide interval, which means the tuning of this hyper-parameter is not difficult. Additionally, we used the same orthogonal loss weight in the Swin Transformer architecture as we did in the ViT architecture. Its performance indicates that our method design is not sensitive to this weight.
>
> | Orth Loss Weight | 0.01  | 0.05  |  0.1  |  0.2  |  0.5  |
> | :--------------: | :---: | :---: | :---: | :---: | :---: |
> | 1-shot Avg mIoU  | 62.59 | 63.01 | 63.22 | 63.18 | 62.87 |
>
>
>
> ## 2. Apply to Swin Transformer
>
> |  Swin-B   |    Stage1     |     Stage2     |      Stage3      |      Stage4      |
> | :-------: | :-----------: | :------------: | :--------------: | :--------------: |
> |   shape   | H/4 * W/4 * C | H/8 * W/8 * 2C | H/16 * W/16 * 4C | H/32 * W/32 * 8C |
> | layer num |       2       |       2        |        18        |        2         |
>
> Thank you for your suggestion. We further validated the effectiveness of the method on Swin Transformer. Since the layers in Swin Transformer do not reside in the same feature space, two additional steps are required: 1) For features from stage 2 to stage 4, we upsample them spatially to H/4*W/4; 2) For features from stage 1 to stage 3, we add three mapping linear layers (c,8c), (2c, 8c), and (4c, 8c) to map them to the same feature space as that of stage 4. The mapping layers are trained alongside the model during the source domain training phase. The performance results are as follows: our method is well-suited to Swin Transformer, and Swin Transformer shows a significant improvement in performance compared to ViT. Thank you again for your suggestion!
>
> |    1-shot     |  FSS1000  | Deepglobe |   ISIC    |  ChestX   | Mean      |
> | :-----------: | :-------: | :-------: | :-------: | :-------: | --------- |
> |     ViT-B     |   77.80   |   33.18   |   36.99   |   51.54   | 49.88     |
> |    Swin-B     |   79.85   |   37.24   |   39.90   |   66.73   | 55.93     |
> | Swin-B + Ours | **81.02** | **46.63** | **49.19** | **83.85** | **65.17** |

---

### Official Review · Reviewer_oMi8 · 2025-03-14

**Overall Recommendation:** 4

**Summary:**

This paper addresses feature entanglement in Cross-Domain Few-Shot Segmentation (CD-FSS), discovering that ViT features assign equal weights to both meaningful and meaningless pattern matches when comparing images. To solve this, the authors propose a self-disentanglement and re-composition framework with three modules: OSD to reduce feature correlation, CPC for pattern re-composition, and AFW for target-domain adaptation. Their approach outperforms state-of-the-art methods by 1.92% and 1.88% in 1-shot and 5-shot settings respectively.

**Claims And Evidence:**

The claims in this submission are generally well-supported by evidence.

**Essential References Not Discussed:**

In disentangled representation methods, there is a lack of the Slot attention-based method. [1, 2, 3]
[1] Locatello, Francesco, et al. "Object-centric learning with slot attention." Advances in neural information processing systems 33 (2020): 11525-11538.
[2] Seitzer, Maximilian, et al. "Bridging the gap to real-world object-centric learning." arXiv preprint arXiv:2209.14860 (2022).
[3] Chen, Chaoqi, Luyao Tang, and Hui Huang. "Reconstruct and Match: Out-of-Distribution Robustness via Topological Homogeneity." Advances in Neural Information Processing Systems 37 (2024): 125588-125607.

**Experimental Designs Or Analyses:**

All experiments use common benchmarks in the CD-FSS field. The experimental design looks rational, no significant issues.

**Methods And Evaluation Criteria:**

Yes

**Other Comments Or Suggestions:**

In the related work section, should discuss some studies based on Slot Attention (SA) representations, since SA's motivation is also about disentangled and then reconstructing.

**Other Strengths And Weaknesses:**

Pros:
1. The paper tackles disentangled representations, which is an important research area. It's the first to use disentangled features for CD-FSS tasks.
2. Well-organized and easy to follow, with clear figures.
3. They propose a novel pipeline, explain their motivation clearly, and back it up with experiments that validate their approach.
4. Experiments are thorough and show excellent performance.

Cons:
1. The paper assumes that comparing different layers of ViT is mostly meaningless. But in reality, ViT layers work together by dynamic self-attention: shallow layers focus on details (like edges), while deeper layers handle the big picture (like the whole bird). In Fig. 4, The CKA metric used in the paper only checks if features distribution similarity, but it misses how these layers complement each other. Also, in L197-L198, the The authors also acknowledge that dynamic self-attention mechanisms may introduce certain meaningful cross-layer associations.
2. In Tab. 1, the CKA value for Top-12 Avg. (shifted matches) (0.8126) is significantly higher than that of Layer-wise Avg. (diagonal matches, 0.6107), suggesting that some cross-layer comparisons might be more effective. However, the authors did not exploit this observation to optimize their method; instead, they directly suppressed non-diagonal comparisons via the OSD module.
3. The author did not compare with existing disentangled representations methods [1, 2].

[1] Locatello, Francesco, et al. "Object-centric learning with slot attention." Advances in neural information processing systems 33 (2020): 11525-11538.

[2] Seitzer, Maximilian, et al. "Bridging the gap to real-world object-centric learning." arXiv preprint arXiv:2209.14860 (2022).

**Questions For Authors:**

The paper mentions a low parameter count, but how does it perform in terms of FPS and overall time efficiency?

**Relation To Broader Scientific Literature:**

**CD-FSS**: Extends the benchmark established by PATNet (Lei et al., 2022), addressing the feature entanglement problem in existing methods (e.g., PFENet, HSNet) to improve cross-domain transferability.

**ViT Structural Decomposition**: Builds on ViT interpretability studies (Gandelsman et al., 2023), proposing the first application of ViT’s residual stream decomposition for feature disentanglement—unlike classical approaches (e.g., InfoGAN), it requires no auxiliary networks.

**Theoretical Claims:**

N/A - This is an experimental paper with no theoretical claims or proofs that require verification.

---

> ### Author Rebuttal · Authors · 2025-04-01
>
> ## 1. Compare with more disentangle-based methods and Discussion on slot attention
>
> Our approach differs from slot attention-based methods [1, 2] in two fundamental aspects: 1) Slot attention primarily disentangles distinct objects through object-centric representation optimization, exhibiting coarser granularity, whereas our method focuses on disentangling different patterns within the same object at a finer granularity level. 2) While slot attention employs additional iterative attention modules (external networks) for disentanglement, we leverage the inherent property of ViT layers that naturally attend to distinct spatial regions, augmented with orthogonality constraints to reinforce semantic separation, without requiring additional networks. To validate our method's effectiveness, we conduct comparative evaluations with two representative slot attention-based disentanglement approaches [1, 2]：
>
> |  1-shot  |  FSS1000  | Deepglobe |   ISIC    |  ChestX   | Mean      |
> | :------: | :-------: | :-------: | :-------: | :-------: | --------- |
> | Baseline |   77.80   |   33.18   |   36.99   |   51.54   | 49.88     |
> |   [1]    |   79.05   |   37.83   |   41.22   |   69.59   | 56.92     |
> |   [2]    |   79.62   |   38.57   |   40.89   |   72.33   | 57.85     |
> |   Ours   | **80.31** | **43.15** | **46.57** | **82.86** | **63.22** |
>
> We promise to add the discussion on Slot Attention in the final version for both the related work and experiments.
>
> [1] Locatello, Francesco, et al. "Object-centric learning with slot attention." Advances in neural information processing systems 33 (2020): 11525-11538.
>
> [2] Seitzer, Maximilian, et al. "Bridging the gap to real-world object-centric learning." arXiv preprint arXiv:2209.14860 (2022).
>
> ## 2. Misunderstanding of our cross-comparison design
>
> We would like to point out that we did not deny the effectiveness of cross-comparison of layers. Instead, **we surely acknowledge the importance of cross-comparisons, and our method is majorly built upon cross-comparisons**. Specifically,
>
> (1) We hold that different layers tend to focus on different patterns that are complementary to each other. Our analysis is based on this characteristic, utilizing it to achieve self-disentanglement, and the complementary of patterns is achieved by the recomposition. The comparison of feature-similarity distributions measured by CKA is intended to demonstrate this dynamic nature, **indicating that different layers may still have meaningful connections, making it possible to complement different layers by the cross-comparisons, instead of denying it**.
>
> (2) It is because of the meaningful connections across different layers that **we design AFW to dynamically assign weights for these cross-comparisons for recomposition**, instead of simply forcing the position-wise comparison that ignores the cross comparison. In our experiments (Fig.9 and Tab.5), we also validated the effectiveness of the cross comparison.
>
> (3) Also, as validated in Fig.9, our **OSD does not suppress non-diagonal comparisons**; its role is to encourage layers to focus on more complementary patterns instead of those repeated ones that have been focused on by other layers. With the AFW module, our model still encourages the cross-comparisons across layers, while the complementary of patterns is enhanced by the OSD module.
>
> ## 3. Computational efficiency
>
> We compared our method with PATNet, HSNet, and SSP. Our method demonstrated higher computational efficiency compared to the other methods. This is because we do not require additional networks but instead leverage ViT’s inherent structure for feature separation.
>
> |                   | PATNet | HSNet |  SSP  |   Ours    |
> | ----------------- | :----: | :---: | :---: | :-------: |
> | FLOPs (G)         | 22.63  | 20.12 | 18.97 | **18.86** |
> | Training Time (h) |  6.32  | 5.61  | 5.12  | **5.07**  |

---

### Official Review · Reviewer_PmpS · 2025-03-14

**Overall Recommendation:** 4

**Summary:**

The paper addresses the feature entanglement problem in Cross-Domain Few-Shot Segmentation (CD-FSS) by leveraging the structural decomposition of Vision Transformers (ViTs). The authors identify that cross-layer comparisons in ViTs entangle meaningful and irrelevant patterns, leading to reduced transferability. To resolve this, they propose three modules: (1) Orthogonal Space Decoupling (OSD) to disentangle features via orthogonal constraints, (2) Cross-Pattern Comparison (CPC) to enable adaptive cross-layer feature matching, and (3) Adaptive Fusion Weight (AFW) for target-domain fine-tuning. Experiments on four CD-FSS datasets demonstrate that the proposed approach outperforms state-of-the-art methods.

**Claims And Evidence:**

The claims are well-supported by empirical evidence:
1. Feature entanglement: Validated via Centered Kernel Alignment (CKA) analysis, showing lower domain similarity for cross-layer vs. layer-wise matches (Fig. 4, Table 1).
2. Module effectiveness: Ablation studies confirm the contributions of OSD, CPC, and AFW (Tables 3–7).
3. Superior performance: Results on diverse datasets (Table 2) and visualizations (Figs. 6–9) substantiate improvements.

**Essential References Not Discussed:**

The paper includes most relevant references, but it would be beneficial to cite works that analyze ViT feature disentanglement and its impact on domain generalization.

**Experimental Designs Or Analyses:**

The experiments are well-designed and cover multiple aspects:
1. Comparison with SOTA on four CD-FSS datasets, demonstrating superior generalization ability.
2. Ablation studies to verify the contribution of each module (OSD, CPC, AFW).
3. Comparison of different distance metrics (Euclidean, Cosine, EMD, Dot Product).
4. Feature visualization to support the claim that ViT decomposition enables disentanglement.

Overall, the experimental methodology is robust and comprehensive.

**Methods And Evaluation Criteria:**

Methods: The ViT decomposition and proposed modules (OSD, CPC, AFW) are conceptually sound, leveraging ViT’s residual structure for disentanglement. The use of orthogonal constraints and adaptive weighting aligns with the goal of reducing feature entanglement.

Evaluation: Standard CD-FSS benchmarks and metrics (mIoU) are appropriate. The inclusion of both natural and medical imaging datasets strengthens validity.

**Other Comments Or Suggestions:**

The method section is dense; a flowchart or pseudocode could improve readability.

**Other Strengths And Weaknesses:**

Strengths:
1. Novel perspective: The paper introduces a unique way of handling feature entanglement in ViTs for CD-FSS.
2. Methodological soundness: The proposed modules (OSD, CPC, AFW) are well-justified and experimentally validated.
3. Strong experimental results: The method consistently outperforms SOTA across multiple datasets and settings.
4. Computational efficiency: Unlike decoder-based architectures like APSeg, the proposed method only requires an encoder, making it more efficient.

Weaknesses:
1. Theoretical justification of ViT decomposition: While the empirical results support the method, a more formal theoretical analysis of why ViT decomposition is optimal for disentanglement could be provided.
2. The author's method, limited to ViT, may be outdated. Its applicability to more recent architectures like Swin Transformer is questionable. This is something I care about deeply. If the author could answer this question, I would be very willing to significantly improve my score.
3. Limited discussion on broader applicability: The method is focused on CD-FSS, but could it benefit general few-shot segmentation tasks?
4. Limited discussion on computational efficiency (e.g., inference speed) are not deeply analyzed.

**Questions For Authors:**

No questions.

**Relation To Broader Scientific Literature:**

The paper is well-connected to related work in CD-FSS and feature disentanglement.
1. Compared to existing CD-FSS approaches (e.g., PATNet, APSeg, DRA, ABCDFSS), this work introduces a novel ViT decomposition-based perspective.
2. Compared to disentanglement methods (e.g., InfoGAN, Disentangled-VAE, DFR), the proposed method does not require additional networks but instead leverages ViT’s inherent structure for feature separation.

However, additional discussion on ViT feature disentanglement in other contexts (e.g., domain generalization) could further strengthen the paper.

**Theoretical Claims:**

The paper does not present formal theoretical proofs but provides a mathematical interpretation of ViT decomposition (Eqs. 1–7) and entanglement analysis. The hypothesis about cross-layer comparisons is empirically validated through CKA and ablation studies. The derivations are logically consistent and align with the experimental results.

---

> ### Author Rebuttal · Authors · 2025-04-01
>
> ## 1. The theoretical analysis of the effectiveness of ViT disentanglement
>
> In cross-domain few-shot segmentation tasks, models need to transfer knowledge from the **source domain $\mathcal{S}$** with abundant annotations to the **target domain $\mathcal{T}$** with limited data. Let $\mathcal{H}$ represent the hypothesis space of the segmentation model. The upper bound of the generalization error for target domain risk $\epsilon_{\mathcal{T}}(h)$ is defined as:
> $$
> \epsilon_{\mathcal{T}}(h) \leq \epsilon_{\mathcal{S}}(h) + d_{\mathcal{H}}(\mathcal{S}, \mathcal{T}) + \lambda,
> $$
> where $h$ denotes features extracted by the encoder, $\epsilon_{\mathcal{S}}(h)$ is the source domain risk, $d_{\mathcal{H}}(\mathcal{S}, \mathcal{T})$ represents the $\mathcal{H}$-divergence (domain gap) between the source and target domains, and $\lambda$ is the irreducible ideal joint risk.
>
> Our approach reduces $\epsilon_{\mathcal{T}}(h)$ through two mechanisms:
>
> 1）**Adaptive Fusion Weights (AFW):** adaptively assigns higher weights to semantically appropriate matches, leading to better alignment (as confirmed by the experiments on the source domain in Answer 3), thereby optimizing source domain output and reducing $\epsilon_{\mathcal{S}}(h)$.
>
> 2）**Domain-Invariant Component Isolation:** Minimizing $d_{\mathcal{H}}(\mathcal{S}, \mathcal{T})$ by isolating domain-invariant patterns (e.g., object parts) via:
> $$
> d_{\mathcal{H}}(\mathcal{S}, \mathcal{T}) \approx \sum_{i=1}^L \sum_{j=1}^L w_{ij}d_{\mathcal{H}}^{(ij)}(\mathcal{S}, \mathcal{T}) \leq \sum_{i=1}^L \sum_{j=1}^L d_{\mathcal{H}}^{(ij)}(\mathcal{S}, \mathcal{T})
> $$
> where $d_{\mathcal{H}}^{(ij)}$ denotes the inter-layer domain discrepancy. By leveraging the self-disentangle property and the orthogonal constraints from the OSD module, inappropriate matches are learned to have small $w_{ij}$, reducing the inter-layer mutual information $\mathbb{I}[\mathbf{h}_i \mathbf{h}_j^T]$, thereby tightening the boundary.
>
> ## 2. Effectness in Swin Transformers
>
> |  Swin-B   |    Stage1     |     Stage2     |      Stage3      |      Stage4      |
> | :-------: | :-----------: | :------------: | :--------------: | :--------------: |
> |   shape   | H/4 * W/4 * C | H/8 * W/8 * 2C | H/16 * W/16 * 4C | H/32 * W/32 * 8C |
> | layer num |       2       |       2        |        18        |        2         |
>
> Thank you for your suggestion. We further validated the effectiveness of the method on Swin Transformer. Since the layers in Swin Transformer do not reside in the same feature space, two additional steps are required: 1) For features from stage 2 to stage 4, we upsample them spatially to H/4*W/4; 2) For features from stage 1 to stage 3, we add three mapping linear layers (c,8c), (2c, 8c), and (4c, 8c) to map them to the same feature space as that of stage 4. The mapping layers are trained alongside the model during the source domain training phase. The performance results are as follows: our method is well-suited to Swin Transformer, and Swin Transformer shows a significant improvement in performance compared to ViT. Thank you again for your suggestion!
>
> |    1-shot     |  FSS1000  | Deepglobe |   ISIC    |  ChestX   |   Mean    |
> | :-----------: | :-------: | :-------: | :-------: | :-------: | :-------: |
> |     ViT-B     |   77.80   |   33.18   |   36.99   |   51.54   |   49.88   |
> |    Swin-B     |   79.85   |   37.24   |   39.90   |   66.73   |   55.93   |
> | Swin-B + Ours | **81.02** | **46.63** | **49.19** | **83.85** | **65.17** |
>
> ## 3. Our methods can benefit general few-shot segmentation tasks and domain generalization tasks
> We measure the FSS performance of our methods on Pascal. Pascal consists of 20 classes and is set to a 4-fold configuration in the FSS setup. This means training is conducted on 5 classes, while testing is performed on 15 classes that were not seen during the training phase. The experimental results show that our method can also effectively improve the performance of general FSS tasks.
> |1shot|Fold0|Fold1|Fold2|Fold3|Mean|
> | ------- | :------: | :------: | :------: | :------: | :------: |
> | Baseline|61.5|68.2|66.7|52.5|62.2|
> | Ours |**63.1**|**70.3**|**67.8**|**55.4**|**64.2**|
>
> Under the domain generalization setting, our method, trained on Pascal and tested on FSS1000, demonstrates an improvement in domain generalization.
> |    | baseline |ours|
> | --- | :-----: | --- |
> |FSS|72.6|75.3|
>
> ## 4. Discussion on computational efficiency
> We compared our method with PATNet, HSNet, and SSP. Our method demonstrated higher computational efficiency compared to the other methods. This is because we do not require additional networks but instead leverage ViT’s inherent structure for feature separation.
>
> |              |PATNet|HSNet|SSP|Ours|
> | ------------ | :----: | :---: | :---: | :-------: |
> | FLOPs (G) |22.63|20.12|18.97|**18.86**|
> | Training Time (h)|6.32|5.61|5.12|**5.07**|
>
> ## 5. Method section
> Thank you for your suggestion; we will work on improving it in the final version.

---

### Decision · Program_Chairs · 2025-05-01

**Decision:**

Accept (poster)

**Comment:**

Dear authors,
Thank you for the submitting draft. This draft received overall positive review, with three weak accept and 1 Accept.
Reviewer oMi8 has indicated to raise the score and 4QAQ has stated to be "satisfied with the authors' rebuttal.".
Taking these comments inconsideration, we are assigning Accept.


regards

AC